# Draft Genome of *Nocardia canadensis* sp. nov. Isolated from Petroleum-Hydrocarbon-Contaminated Soil

**DOI:** 10.3390/microorganisms11122972

**Published:** 2023-12-12

**Authors:** Fahad Alotaibi, Soon-Jae Lee, Zakaria Lahrach, Marc St-Arnaud, Mohamed Hijri

**Affiliations:** 1Institut de Recherche en Biologie Végétale, Université de Montréal, 4101 East Sherbrooke St., Montréal, QC H1X 2B2, Canada; soon-jae.lee@unil.ch (S.-J.L.); zakaria.lahrach.1@umontreal.ca (Z.L.); marc.st-arnaud@umontreal.ca (M.S.-A.); 2Department of Soil Science, King Saud University, Riyadh 11564, Saudi Arabia; 3African Genome Center, University Mohammed VI Polytechnic (UM6P), Ben Guerir 43150, Morocco

**Keywords:** alkane 1-monooxygenase (*alkB*), alkanes, genome sequencing, *Nocardia canadensis*, plant-growth-promoting rhizobacteria

## Abstract

The bacterial strain WB46 was isolated from the rhizosphere of willow plants (*Salix purpurea* L.) growing in soil contaminated with petroleum hydrocarbons. The strain was subjected to whole-genome shotgun sequencing using Illumina HiSeq. Its draft genome is 7.15 Mb, with a 69.55% GC content, containing 6387 protein-coding genes and 51 tRNA and 15 rRNA sequences. The quality and reliability of the genome were assessed using CheckM, attaining an estimated genome completeness of 98.75% and an estimated contamination of 1.68%. These results indicate a high-quality genome (>95%) and low contamination (<5%). Many of these genes are responsible for petroleum hydrocarbon degradation, such as alkane 1-monooxygenase (*alkB*) and naphthalene dioxygenase (*ndo*). 16S rRNA gene analysis, including in silico DNA–DNA hybridization (DDH) and average nucleotide identity (ANI), showed that strain WB46 belongs to the genus *Nocardia*, and the most closely related species is *Nocardia asteroides.* The strain WB46 showed a distance of 63.4% and sequence identity of 88.63%, respectively. These values fall below the threshold levels of 70% and 95%, respectively, suggesting that the strain WB46 is a new species. We propose the name of *Nocardia canadensis* sp. nov. for this new species. Interestingly, the sequence divergence of the 16S rRNA gene showed that the divergence only occurred in the V2 region. Therefore, the conventional V3–V4, V5–V7, or V8–V9 targeting metabarcoding, among others, would not be able to assess the diversity related to this new species.

## 1. Introduction

Intensive industrial activities, such as extracting oil and gas, employing inorganic fertilizer-based agriculture, mining for minerals, and disposing of industrial waste, are associated with risks of environmental contamination, which present a global challenge [1,2]. Of particular concern are petroleum hydrocarbons (PHCs), which can result in high-risk oil spills and environmental contamination in both aquatic and terrestrial ecosystems [3,4]. PHCs, such as crude oil, are heterogeneous organic mixtures composed of carbon and hydrogen atoms, split into two major fractions: aliphatic hydrocarbons (alkenes, alkynes, or alkanes) and aromatic hydrocarbons (including monoaromatic and polycyclic aromatic hydrocarbons (PAHs)) [5]. Most commonly, the sources of PHC contamination are anthropogenic, derived from accidental release (e.g., diesel, fuel, solvents) and industrial activities (e.g., electricity production, petrochemical) [6]. Such environmental contamination with PHC products has caused significant detriment to various ecosystems, including soils, with serious economic consequences [7].

The use of plants and their associated microbes, known as phytoremediation, has been suggested as a promising method for managing PHC pollution in soil [8,9]. This environmentally friendly and solar-powered approach has a small carbon footprint and has been successful in moderately polluted soils [7]. However, it may not be as effective in heavily polluted soils due to hindered plant growth in these conditions. *Salix* spp., commonly found in various habitats in North America, are known for their high tolerance to chronic PHC pollution [2,10]. Therefore, these plants are particularly well-suited for phytoremediation efforts in addressing PHC contamination. Two studies by Alotaibi et al. [2,11] identified and characterized over 400 bacterial isolates from highly polluted soil environments with diverse abilities in degrading PHCs and promoting plant growth. Among the isolated bacteria strains, WB46, belonging to the genus *Nocardia,* was selected for whole-genome shotgun sequencing [11].

The genus *Nocardia* belongs to the family *Nocardiaceae* of the order *Corynebacteriales*, within the phylum *Actinobacteria* [12]. *Nocardia* species are ubiquitous in both aquatic and terrestrial habitats, such as soil, water, and the decaying fecal deposits of animals [13], with predominant importance in clinical and environmental settings [14,15]. Since the first isolation of *Nocardia* sp. by Edmond Nocard in 1888 [16], more than 119 species have been described so far (http://www.bacterio.net/ accessed on 8 November 2023). Many species of *Nocardia* are opportunistic pathogens for humans and animals [17,18]. However, more recently, several species of *Nocardia* were found to produce new bioactive substances [19,20] and to degrade various petroleum hydrocarbon compounds [21]. Evidently, this genus shows the potential to be exploited for the biodegradation of petroleum hydrocarbons. Still, there are only a few species isolated and validated for biodegradation ability. *Nocardia* sp. strain WB46 was isolated from the rhizosphere of willow plants (*Salix purpurea* L.) growing in soil contaminated with petroleum hydrocarbons from the site of a former petrochemical plant located at Varennes, Québec, Canada [2]. This strain was subjected to the whole-genome shotgun sequencing using Illumina HiSeq, as well as in vitro analyses, which indicated that the bacterium can utilize a wide range of petroleum hydrocarbons as the sole source of carbon to grow and reproduce, including aliphatic and polycyclic aromatic hydrocarbons [2]. The strain *Nocardia* sp. WB46 also displayed positive activities for some plant-growth-promoting traits such as phosphate solubilization and siderophores production, when tested in vitro, suggesting it could be a useful partner for bioremediation with plants (Table 1) [2]. An analysis of the 16S rRNA gene, using methods such as in silico DNA–DNA hybridization and average nucleotide identity, was performed on *Nocardia* sp. WB46 and a closely related strain, *Nocardia asteroides*. The results indicated that the strain WB46 is a newly discovered species, which was named as *Nocardia canadensis*.

## 2. Materials and Methods

### 2.1. Isolation and Media Culture of Bacteria

*Nocardia* sp. strain WB46 was isolated from the rhizosphere of willow (*Salix purpurea* L.) growing in soil polluted with petroleum hydrocarbons, as part of a large phytoremediation pilot project [2,22]. Willow plants were grown on a former petrochemical plant located at Varennes, Québec, Canada [22,23], which was in operation from 1953 to 2008 [23]. The soil at the site was contaminated with a mixture of alkanes and polycyclic aromatic hydrocarbons (PAHs), reaching concentrations up to 3590 mg kg^−1^ [22], exceeding the limit for land reuse defined by the government of Québec for industrial areas. *Nocardia* sp. strain WB46 was isolated from the rhizosphere of willow using Bushnell-Haas medium amended with 1% diesel, as the sole carbon source, as described elsewhere [2].

### 2.2. DNA Extraction and Whole-Genome Shot Gun Sequencing

Genomic DNA was extracted from stationary-phase cells grown in 1/10 Trypticase Soy Broth (TSB) (Difco Laboratories, Detroit, MI, USA) medium using the DNeasy UltraClean Microbial Kit (Qiagen, Toronto, Canada), according to the manufacturer’s instructions. DNA concentration was determined on a Qubit fluorometer (Thermo Fisher Scientific, Mississauga, ON, Canada). The genomic library was prepared with an NEB Ultra II kit (New England BioLabs Inc., Ipswich, MA, USA) and sequenced on an Illumina MiSeq platform with 250 bp paired-end chemistry.

### 2.3. Bioinformatics Pipeline and Processing of Data

Raw paired-end sequences were subjected to quality trimming using SeqMan NGen software (Version 12, DNAStar Inc., Madison, WI, USA). Genome assembly was also performed using SeqMan NGen software (Version 12, DNAStar Inc., Madison, USA). Genome completeness was determined using CheckM (V2.1) [24]. Gene annotation was performed using the NCBI Prokaryotic Genome Annotation Pipeline (PGAP) (Tatusova et al., 2016) [25]. The in silico DNA–DNA hybridization (DDH) value was calculated using the Genome-to-Genome distance calculator version 2.1 (GGDC) (http://ggdc.dsmz.de/ggdc_background.php#, accessed on 11 December 2023) [26]. Average nucleotide identity (ANI) analyses were conducted between *Nocardia* sp. strain WB46 and closely related strains using the NCBI’s PGAP–taxcheck option [26]. The 16S rRNA gene sequence (length of 1516 bp) derived from the assembled genome was compared with the available sequences in the Ribosomal Database Project (RDP) using the SeqMatch tool (https://rdp.cme.msu.edu/seqmatch/seqmatch_intro.jsp, accessed on 11 December 2023). The tool used to visualize the circular genome was GenVision Pro (Version 9, DNASTAR, Inc., Madison, WI, USA).

### 2.4. Description of Nocardia canadensis sp. nov.

*Nocardia canadensis* (N.L. masc. adj. *canadensis*, Of Canada, from which this microorganism was isolated): The species name was registered in SeqCode under the Canonical URL, https://seqco.de/i:32942 (accessed on 11 December 2023). On the basis of its morphology and physiology, this bacterium is an aerobic actinobacterium that is gram-positive, non-acid-fast, and non-motile. The colonies it forms on the culture media can range from white to yellow with aerial mycelium-like colonies.

## 3. Results and Discussion

The DNA concentration obtained was 42 ng/µL. In total, 1,605,568 raw paired-end sequences, providing approximately 34-fold coverage of the genome, were subjected to quality trimming using the SeqMan NGen software version 12.0. From assembling, we obtained 7,150,745 bp in 10 contigs. The genome of *Nocardia* sp. strain WB46 has an average G + C content of 69.55% and includes 6387 predicted protein-coding sequences (CDSs), 15 rRNAs (5S, 16S, 23S), 51 tRNAs, and 3 noncoding RNAs (ncRNAs) sequences. Detailed genomic information is presented in Appendix A and in Figure 1.

The quality and reliability of the genome were confirmed using CheckM [24]—with an estimated genome completeness of 98.75% and an estimated contamination of 1.68%—indicating a high-quality genome (>95%) and low contamination (<5%), respectively. In silico DNA–DNA hybridization (DDH), Type (Strain) Genome Server (TYGS), average nucleotide identity (ANI), and 16S rRNA gene analyses all suggested that *Nocardia* sp. strain WB46 is in fact a new species (Table 2). The in silico DNA–DNA hybridization (DDH) value between *Nocardia* sp. strain WB46 and a closely related strain *Nocardia asteroides* showed a distance of 63.4% (Table 2), which is below the threshold level of 70% recommended by [27] for assigning bacterial strains to the same species, thus suggesting that *Nocardia* sp. strain WB46 is a new species. The *Nocardia* sp. strain WB46 was also uploaded to the Type (Strain) Genome Server (TYGS) (https://tygs.dsmz.de, accessed on 11 December 2023) for a whole-genome-based taxonomic analysis. *Nocardia* sp. strain WB46 did not belong to any species found in the TYGS database and was tagged as a potential new species (Table 2). Additionally, the average nucleotide identity (ANI) results predicted *Nocardia* sp. strain WB46 as *Nocardia asteroides* (Table 2), but the value of 88.63% was below the generally proposed species boundary cut-off of 95–96% [28]. Additional pairwise genome comparisons between *Nocardia* sp. strain WB46 and *Nocardia asteroides* with other (ANI)’s tools all suggested *Nocardia* sp. strain WB46 being a new species: ChunLab’s ANI Calculator (https://www.ezbiocloud.net/tools/ani, accessed on 11 December 2023) [29] OrthoANIu = 88.16%, JSpeciesWS (http://jspecies.ribohost.com/jspeciesws/, accessed on 11 December 2023) ANIb = 87.27%, ANIm = 89.20%, and Kostas lab ANI Calculator (http://enve-omics.ce.gatech.edu/ani/, accessed on 11 December 2023) two-way ANI = 88.23%.

The DDH and genome similarity comparison clearly showed that the genome of the novel strain WB46 has diverged from the genome of existing registered strains of *Nocardia*. Nevertheless, since the comparison is only based on the genomes available in the database, we compared the 16S rDNA sequence of WB46 with those present in the Ribosomal Database Project (RDP), which contains sequences of all environmental *Nocardia* strains to further ensure its novel nature. A 1516 bp 16S rDNA sequence was extracted from the assembled genome of strain WB46 and analyzed using the SeqMatch tool. The results showed that the sequence was almost identical to that of a *Nocardia asteroides* at 98.8% similarity (Table 2). Thus, strain WB46 was concluded to belong to the genus *Nocardia.* Recently, 16S rRNA gene sequence similarity threshold values in the range of 98.2–99.0% have been widely accepted and used to differentiate two species [25,28,30], instead of the 97% threshold previously used [31], which supports that *Nocardia* sp. strain WB46 is a potent new species. To understand the evolutionary relatedness of the *Nocardia* sp. strain WB46 with other closely related *Nocardia* species, a phylogenetic analysis was conducted with the complete 16S rRNA gene sequence of the strain WB46. First, BLASTn was performed against the 16S rRNA sequence collection (Bacteria/Archaea) of NCBI. The top 100 hits with E-value above 1 × 10^−100^ and a percent sequence identity above 90% were used to produce a multiple sequence alignment through MUSCLE v3.5, which was then further trimmed utilizing Gblock v0.91b [32]. Finally, a phylogenetic tree was computed, the best nucleotide evolution model was selected using JModelTest2 [33], and the model of GTR + G + I was selected. The Bayesian phylogenetic analyses were conducted using BEAST2.5 [34], with 10,000,000 generations and a burn-in of the first 20% of generations. The resulting phylogeny was visualized using iTOL [35]. Interestingly, *Nocardia* sp. strain WB46 did not cluster with any groups at the species level, showing its 16S rDNA sequence divergence from other publicly available *Nocardia* species in the Genbank database. The Bayesian phylogenetic analysis (Figure 2) suggested that *Nocardia* sp. strain WB46 is phylogenetically closely related species to *N*. *asteroides* (posterior probability 1.0/1.0) but further diverged from the ancestor of *N. asteroides* with high posterior probability support for speciation (0.7/1.0) to form a monophyletic node. Surprisingly, the divergence of sequence in *Nocardia* sp. strain WB46 only occurred in the V2 region at position 108–110 bp and 121 bp in the multiple sequence alignment (Figure 3). It has been suggested that the V3-V4 region of the 16S rRNA is informative for understanding bacterial diversity; thus, it is widely used in ecological and environmental studies [36], even though the value of other regions of variance was also emphasized [37]. The position of informative sequence divergence in our study shows that the V2 region should be taken into account for capturing the diversity of this ecologically important bacterial taxa.

Genes connected with the degradation of petroleum hydrocarbons were found in the genome of *Nocardia* sp. strain WB46 (Appendix A). Alkane 1-monooxygenase (*alkB*) and cytochrome P450 hydroxylase (*CYP153*) are important alkane hydroxylases responsible for microbial aerobic alkane degradation in oil-polluted environments. These enzymes hydroxylate alkanes to alcohols, which are further oxidized to fatty acids and catabolized via the bacterial β-oxidation pathway [39]. Previous studies showed that the gene repertoire of *alkB* and *CYP153* are diverse among species of Nocardia. For instance, *Nocardia cyriacigeorgica* GUH-2 has two copies of *alkB* and also two copies of *CYP153*, while *Nocardioidaceae bacterium* Broad-1 has two copies of *alkB* but only one copy of *CYP153* [40]. It has been reported that many *Actinobacteria* genomes containing *CYP153* genes also had *alkB* genes, implying a potential link between the CYP153 and alkB genes in the *Actinobacteria* [40,41]. Interestingly, the genome of *Nocardia* sp. strain WB46 has no *CYP153* gene but has three copies of *alkB* genes. Therefore, its alkane-degrading capability could be associated mainly with the *alkB* activity. Additionally, two naphthalene dioxygenase (*ndo*) genes responsible for biodegrading polycyclic aromatic hydrocarbons (PAHs) [42] were also present in the genome of *Nocardia* sp. strain WB46. Further genes related to plant-growth-promoting characteristics were also detected, including phosphate solubilization and siderophore utilization (Appendix A). For example, the genome of *Nocardia* sp. strain WB46 contains genes related to acid phosphates, which play an important role in plant growth by scavenging P, under low P concentrations, from organophosphate compounds in the rhizosphere and thus, increasing P availability to plants [43]. The genome sequence data of *Nocardia* sp. strain WB46 will enhance our understanding of the metabolic capabilities of *Nocardia* strains.

## 4. Conclusions

Sequencing of the genome of the newly identified bacterial species *Nocardia canadensis* WB46, which was initially isolated from soil contaminated with petroleum hydrocarbons surrounding the roots of willow plants, has shown that it carries genetic material essential for the degradation of such hydrocarbons. This was confirmed through subsequent in vitro analysis, which observed the strain’s ability to utilize a variety of petroleum hydrocarbons as its only source of carbon. Apart from this, the strain also possesses several other functions likely to contribute to the promotion of plant growth. In order to assess the strain’s feasibility of rhizoremediation of soils polluted with petroleum hydrocarbons, *Nocardia canadensis* WB46 will be tested in the field, potentially unveiling opportunities for biotechnological applications.

## Figures and Tables

**Figure 1 microorganisms-11-02972-f001:**
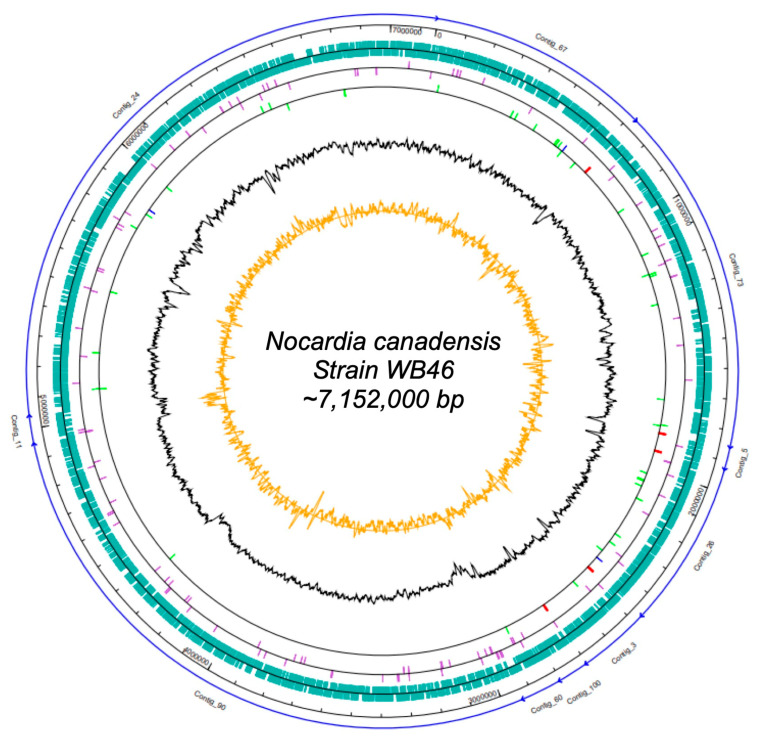
Map of the scaffolded contigs of the *Nocardia* strain WB46. From outer to inner ring: the individual contigs (blue arrows), scale, coding sequences (green) on the forward strand and reverse strand, pseudogenes (purple) on the forward strand and reverse strand, RNA genes on the forward strand and reverse strand (tRNAs green, rRNAs red, other RNAs blue), G + C content (black), and CG-skew (orange).

**Figure 2 microorganisms-11-02972-f002:**
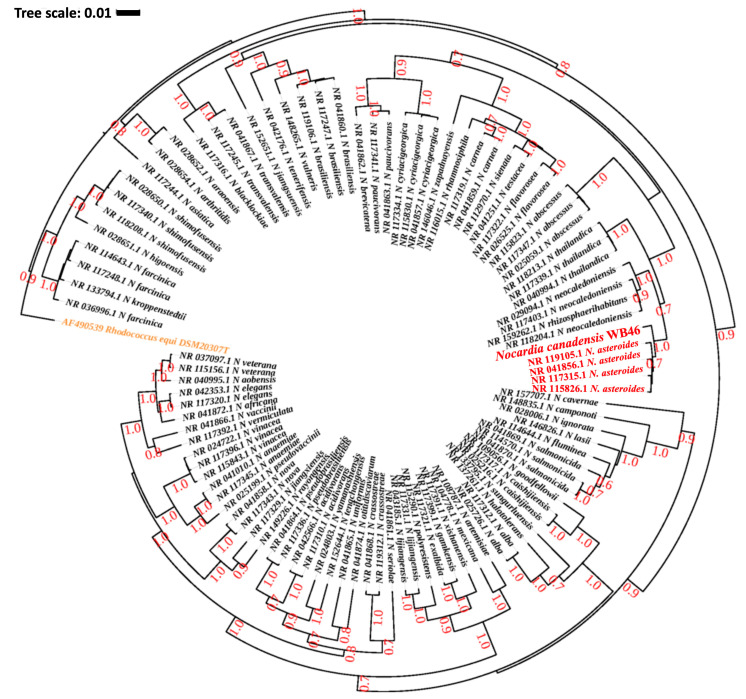
Phylogenetic analysis of *Nocardia* sp. strain WB46 with other species in the *Nocardia* genus, using complete 16S rRNA gene sequences (1358 bp), was conducted using Bayesian phylogenetic analysis. The GTR  +  I + G (with four distinct gamma categories) phylogenetic model showed the lowest BIC value. The tree was rooted using *Rhodococcus equi* as an outgroup (colored orange), following the previous publication of *Nocardia* phylogeny [38]. The numbers at branches correspond to Bayesian posterior probabilities. The branches of a clade (*N. asteroides*), which are suggested to share the most direct common ancestor with *Nocardia* sp. WB46 (colored red) with 1.0/1.0 posterior probability, are colored red.

**Figure 3 microorganisms-11-02972-f003:**
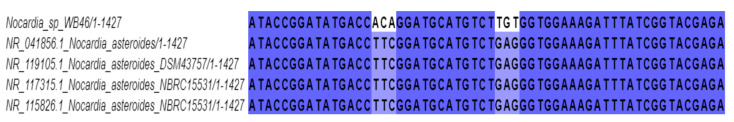
Region of 16S rDNA sequence divergence in multiple sequence alignment of *Nocardia* sp. strain WB46 and *Nocardia asteroides* isolates was observed. Among the full-length 16S rDNA sequence, 6 nucleotide sequences differed in *Nocardia* sp. strain WB46 and the other *Nocardia asteroides* isolates with no sequence divergence. The diverged sequences are shown without a background color.

**Table 1 microorganisms-11-02972-t001:** Hydrocarbon degradation potential and plant-growth-promoting traits of bacterial strain *Nocardia canadensis* WB46 ^a^.

Traits	Assays	Activity
Hydrocarbon degradation potential ^b^	Naphthalene	++
	Phenanthrene	+++
	Pyrene	++
	Dodecane	++
	Hexadecane	+++
Catabolic genes ^c^	Alkane monooxygenase (alkB)	+
	Cytochrome P450 hydroxylase (CYP153)	−
	Naphthalene dioxygenase (nah1)	+
Cell growth measurement at 600 nm ^b^	1% diesel	++++
	1% hexadecane	++++
	2% hexadecane	++++
	3% hexadecane	++++
Plant-growth-promoting traits	1-Aminocyclopropane-1-carboxylate deaminase (ACCD)	−
	Phosphate solubilization	−
	Siderophore production	+ (8.2%)
	Nitrogen fixation	−
	Indole-3-acetic (IAA) production	+ (1.46 μg mL^−1^)
	Ammonia production	+ (2.9 μmol mL^−1^)
Root elongation assay (cm) ^d^	0%	13.4
	1%	12.2
	2%	10.1
	3%	8.6

^a^ This bacterial strain was isolated from the rhizosphere of *Salix purpurea* L. plants that were growing in soil contaminated with petroleum hydrocarbons [2]. For further details regarding the plant-growth-promotion characteristics of this isolate, please refer to [11]. ^b^ The growth capability of bacterial strain with hydrocarbons as the sole carbon and energy source is indicated on 1% (*v*:*v*) hydrocarbon in MS medium, measured by optical density at 600 nm after 1 week of incubation at 28 °C and is rated as ++++, +++, ++, +, and −, from strong to weak growth, respectively. ++++, strong growth (OD600 > 1); +++, growth (OD600 > 0.6); ++, growth (0.6 > OD600 > 0.2); +, growth (OD600 < 0.2); and −, no growth. ^c^ The PCR products for the catabolic genes *alkB*, CYP153, and *nah1* are indicated as “+” for their presence and “−” for their absence. ^d^ % denotes the impact of WB46 treatment on the root length (in cm) of canola plants after seven days of growth in the presence of various concentrations of *n*-hexadecane under growth pouch conditions.

**Table 2 microorganisms-11-02972-t002:** Summary of the results of in silico methods for genome-to-genome comparisons.

Analysis	Value	Comments
In silico DNA–DNA hybridization (DDH)	63.4%	Below the threshold level of 70%
Type (Strain) Genome Server (TYGS)	No match	Tagged as novel species
Average nucleotide identity (ANI)	88.63%	Below the species boundary cut-off of 95–96%
16S rRNA gene sequence (RDP)	98.8%	High similarity to *Nocardia asteroides*

## Data Availability

This whole-genome shotgun project has been deposited at DDBJ/ENA/GenBank under the accession JAVCWB000000000. The version described in this paper is version JAVCWB010000000.

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
