# Peer review of "Draft Genome of Nocardia canadensis sp. nov. Isolated from Petroleum-Hydrocarbon-Contaminated Soil"

_microorganisms, 2023, doi:10.3390/microorganisms11122972_

Round 1
Reviewer 1 Report
Comments and Suggestions for Authors
1- Please, make the title of the manuscript shorter. It is too long.
2- Arrange the keyword on the basis of alphabetic order, and no abbreviations should be used in keywords.
3- The introduction is too short, the authors should use more references especially from recent published articles. Also, authors should pay attention to paragraphing. Each paragraph should start with new contents and words.
4- The format of Tables should be double check, it seems the format of tables are not on the basis of journal s format, for example Table 1.
5- Material and Methods is not clear. Authors should explain and illustrate about the experiments more clearly.
6- Like Introduction and material and methods, the Results and Discussion part is not complete at all.
7- Conclusion has not written very well, and it should be re-written and re-organized completely.
8- The format of References are not on the basis of the format of journal.
9- DOI of all references should be added in Reference list.
10- The English language of the manuscript need Minor revision. There are few grammatical error and the meaning of some sentences are not clear.
Author Response
Comments and Suggestions for Authors
- Please, make the title of the manuscript shorter. It is too long.
Response: The title was shortened as per your suggestion
- Arrange the keyword on the basis of alphabetic order, and no abbreviations should be used in keywords.
Response: The keywords were arranged in alphabetic order.
- The introduction is too short, the authors should use more references especially from recent published articles. Also, authors should pay attention to paragraphing. Each paragraph should start with new contents and words.
Response: Two new paragraphs were added to expand the introduction section. The paragraphing was also checked
- The format of Tables should be double check, it seems the format of tables are not on the basis of journal s format, for example Table 1.
Response: Tables were reformatted.
- Material and Methods is not clear. Authors should explain and illustrate about the experiments more clearly.
Response: The section of Material and Methods was reorganised by subsections and additional information was added.
- Like Introduction and material and methods, the Results and Discussion part is not complete at all.
Response: The section of Results and Discussion was reorganised according the reviewer’s comments
- Conclusion has not written very well, and it should be re-written and re-organized completely.
Response: The conclusion section was rephrased to reflect the study outcomes.
- The format of References are not on the basis of the format of journal.
Response: References were reformatted according to the journal style.
- DOI of all references should be added in Reference list.
Response: Added
- The English language of the manuscript need Minor revision. There are few grammatical error and the meaning of some sentences are not clear.
Response: The English was checked using a professional service.
Reviewer 2 Report
Comments and Suggestions for Authors
Dear authors,
I have a few comments about your paper. First of all, you should clearly state the purpose and novelty of your research.
1. (lines 37-38) more than 119 species have been described so far (http://www.bacterio.net/)
Please specify accession date
2. Table 1. How did you estimate the growth of the culture on hydrocarbons? Did you do serial dilutions and seeding on plates for abundance counts, or analyze it in some other way? Please specify. Also clarify how many solid hydrocarbons (naphthalene, phenanthrene, etc.) you added during cultivation.
3. (lines 75-76) DNA concentration was determined on a Qubit fluorometer
Here or in the Results section, please indicate the concentration (ng/µl) of the DNA preparation obtained
4. For all software listed in the Methods section, please provide the version, and for web services, please provide the accession date.
5. Figure 1. Please specify the software used to create the genomic map
6. (lines 91-92) Please specify the result of genome verification with CheckM (completeness, contamination).
7. It has been reported that almost all the Actinobacteria genomes containing CYP153 genes also had alkB genes, implying a potential link between the CYP153 and alkB genes in the Actinobacteria
Not a fact. For example, representatives of Gordonia alkanivorans have no alkB genes, but do contain CYP153 genes.
10.1128/MRA.01450-19
10.3390/ECP2023-14689
8. (lines 185-186)
It would be interesting to assess the relatedness of these genes to each other. Also, please describe their immediate environment. For example, alkB1 in rhodococci is accompanied by 2 rubredoxin genes and rubredoxin reductase, alkB2 is accompanied only by rubredoxins without reductase. What is it like in your strain?
9. There are a number of typos in the text that require correction. For example, «Nocardia asteroids» instead of Nocardia asteroides.
Author Response
I have a few comments about your paper. First of all, you should clearly state the purpose and novelty of your research.
- (lines 37-38) more than 119 species have been described so far (http://www.bacterio.net/)
Please specify accession date
Response: Done. (Accessed on 08/11/2023)
- Table 1. How did you estimate the growth of the culture on hydrocarbons? Did you do serial dilutions and seeding on plates for abundance counts, or analyze it in some other way? Please specify. Also clarify how many solid hydrocarbons (naphthalene, phenanthrene, etc.) you added during cultivation.
Response: Cell growth were estimated by measuring cells density at 600nm and compared with a control containing no carbon source.
80 µL of naphthalene, phenanthrene, or pyrene, prepared from (5 g L−1) stock solution, was added.
More information regarding the growth and measurement of the culture on hydrocarbons are listed inAlotaibi et al. 2021b and Alotaibi et al. 2022, which are included in the final version of the manuscript.
- (lines 75-76) DNA concentration was determined on a Qubit fluorometer
Here or in the Results section, please indicate the concentration (ng/µl) of the DNA preparation obtained
Response: Done. The DNA concentration was 42 ng/µl.
- For all software listed in the Methods section, please provide the version, and for web services, please provide the accession date.
Response: SeqMan NGen (Version 12, DNASTAR, INC. Madison, WI. USA)
- Figure 1. Please specify the software used to create the genomic map
Response: The tool used to visualize the circular genome was GenVision Pro (Version 9, DNASTAR, INC. Madison, WI. USA).
- (lines 91-92) Please specify the result of genome verification with CheckM (completeness, contamination).
Response: The quality and reliability of the genome was verified using CheckM, determining the estimated genome completeness at 98.75% and the estimated contamination at 1.68%, which represents a high-quality genome (>95%) and low contamination (<5%), respectively [6].
- It has been reported that almost all the Actinobacteria genomes containing CYP153 genes also had alkB genes, implying a potential link between the CYP153 and alkB genes in the Actinobacteria
Not a fact. For example, representatives of Gordonia alkanivorans have no alkB genes, but do contain CYP153 genes.
10.1128/MRA.01450-19
10.3390/ECP2023-14689
Response: Done. Changed almost all to many…
- (lines 185-186)
It would be interesting to assess the relatedness of these genes to each other. Also, please describe their immediate environment. For example, alkB1 in rhodococci is accompanied by 2 rubredoxin genes and rubredoxin reductase, alkB2 is accompanied only by rubredoxins without reductase. What is it like in your strain?
Response: Not applicable.
- There are a number of typos in the text that require correction. For example, «Nocardia asteroides» instead of Nocardia asteroides.
Response: Done.
Reviewer 3 Report
Comments and Suggestions for Authors
This communication indicates that strain WB46 will be described as a new species by the authors (132-133). The authors emphasize in the title and introduction the petrochemical history of the soil. The extent of hydrocarbon pollution of the soil is not noted in the manuscript, however, but it is of secondary relevance in this genome study. The text refers to “many genes” of hydrocarbon degradation but only three are mentioned in the manuscript.
1. 45-47. Nocardia WB46 is not mentioned specifically in the reference Alotaibi et al. 2021.
2. Table 1 footnote. Alotaibi et al. 2021b and Alotaibi et al. 2022 are not listed in the Refences section. These missing references may be important if they have more information on WB46 traits and genes.
3. The manuscript has minor typos (commas, character spacing, syntax, lower/upper case letters), which should be proofed and corrected for the final version.
4. Table 1 lists alkB and nah1, which encode a 1-monooxygenase and an upper pathway enzyme of naphthalene degradation, respectively. Which specific enzyme is nah1? Two genes are not convincing evidence that WB46 has potential for hydrocarbon remediation.
5. Abstract. 19-20. “…many genes responsible for petroleum hydrocarbon degradation...” However, only three genes were detected: ndo, alkB, and nah1. These genes do not suggest the presence of complete degradative pathways.
6. 48-50. “…wide range of petroleum hydrocarbons as the sole source of carbon…” Reference?
7. Footnote to Table 1. …strain is isolated…> strain was isolated. Explain the -, +, ++, and +++ codes. Are they explained in a previous publication?
8. Conclusions. Define PHC, or do not use the abbreviation.
9. The format in the references section varies, especially with journal titles. The format must be standardized.
10. Table S1. Why is the pseudogenes entry in red?
11. Table S3. Include the gene designations. Re-phrase the caption by avoiding abbreviations PHC and PGP. Include the proper gene designations. The “similar to” column, what does it mean? Are the sequences identical? Percent similarity? Be specific.
12. Table S3. The significance of acid phosphatase needs some explanation because most prokaryotes have acid phosphatases. Are most bacteria capable of solubilizing phosphate from other molecules?
Comments on the Quality of English LanguageTypos and syntax errors should be corrected.
Author Response
- 45-47. Nocardia WB46 is not mentioned specifically in the reference Alotaibi et al. 2021.
Response: Nocardia WB46 is mentioned in Alotaibi et al. 2021b which is included in the final version of the manuscript.
- Table 1 footnote. Alotaibi et al. 2021b and Alotaibi et al. 2022 are not listed in the Refences section. These missing references may be important if they have more information on WB46 traits and genes.
Response: Indeed. Alotaibi et al. 2021b and Alotaibi et al. 2022 are important as they have more information on WB46 traits and genes. These two references are included in the final version of the manuscript.
- The manuscript has minor typos (commas, character spacing, syntax, lower/upper case letters), which should be proofed and corrected for the final version.
Response: Done. Corrected.
- Table 1 lists alkB and nah1, which encode a 1-monooxygenase and an upper pathway enzyme of naphthalene degradation, respectively. Which specific enzyme is nah1? Two genes are not convincing evidence that WB46 has potential for hydrocarbon remediation.
Response: Soils are contaminated mainly by two fractions of petroleum hydrocarbons, namely polycyclic aromatic hydrocarbon and aliphatic hydrocarbon fractions of oil. Therefore, the presence of genes, such as ndo, alkB, and nah1, related to degradation of these two fractions in the genome of WB46 indicates its potential for hydrocarbon remediation
- Abstract. 19-20. “…many genes responsible for petroleum hydrocarbon degradation...” However, only three genes were detected: ndo, alkB, and nah1. These genes do not suggest the presence of complete degradative pathways.
Response: ndo, alkB, and nah1 genes are responsible for the degradation of the polycyclic aromatic hydrocarbon and aliphatic hydrocarbon fractions of oil. These two fractions represent major portion of petroleum hydrocarbons. Therefore, the presence of these genes indicates the potential for hydrocarbon remediation.
- 48-50. “…wide range of petroleum hydrocarbons as the sole source of carbon…” Reference?
Response: Done.
- Footnote to Table 1. …strain is isolated…> strain was isolated. Explain the -, +, ++, and +++ codes. Are they explained in a previous publication?
Response: Yes, they were explained in our previous publications Alotaibi et al. 2021b and Alotaibi et al. 2022. These two specific references are included in the final version of the manuscript.
- Conclusions. Define PHC, or do not use the abbreviation.
Response: Done.
- The format in the references section varies, especially with journal titles. The format must be standardized.
Response: Done.
- Table S1. Why is the pseudogenes entry in red?
Response: Done. Corrected.
- Table S3. Include the gene designations. Re-phrase the caption by avoiding abbreviations PHC and PGP. Include the proper gene designations. The “similar to” column, what does it mean? Are the sequences identical? Percent similarity? Be specific.
Response: Done.
- Table S3. The significance of acid phosphatase needs some explanation because most prokaryotes have acid phosphatases. Are most bacteria capable of solubilizing phosphate from other molecules?
Response: Indeed. Many bacteria were able to solubilize P-bound to soil particles through secretion of acid phosphatases with high affinatiy compared with other prokaryote counter partners.
Reviewer 4 Report
Comments and Suggestions for Authors
The present manuscript is really weak in the result part, the strain is obviously a new Nocardia species close to N. asteroides. Authors need to re-analyze the experiment data and propose the new species to meet the journal standard. In addition, the phylogenetical analyse should be improved based on the genome sequence.
Author Response
The present manuscript is really weak in the result part, the strain is obviously a new Nocardia species close to N. asteroides. Authors need to re-analyze the experiment data and propose the new species to meet the journal standard. In addition, the phylogenetical analyse should be improved based on the genome sequence.
Response: We are appreciative of the reviewer's useful feedback that helped us improve the manuscript by completing multiple sections, such as the Results and Discussion. Additionally, we registered the newly isolated strain Nocardia canadensis on the SeqCode platform, thus giving it an official name (Nocardia canadensis (N.L. masc. adj. canadensis, Of Canada, from which this microorganism was isolated). For verification purposes, the data was reanalyzed in detail and additional information was included to enhance the strength of the manuscript.
Round 2
Reviewer 1 Report
Comments and Suggestions for Authors
The article has been revised very well. But, still the article needs Minor revision:
(1) References are not organized and written on the basis of journal s format, for example, the name of journals should be abbreviated.
(2) DOI of all references and articles should be added in References.
(3) Please, delete Salix in keywords, and write the full name of PGPR in the keywords section.
(4) If the article has two corresponding article according to what authors have mentioned by adding two emails, ONE STAR ( * ) should be added for the second corresponding author as well.
After these two Minor Revision, the article can be accepted for publication.
Author Response
Answers point-by-point:
The article has been revised very well. But, still the article needs Minor revision:
- References are not organized and written on the basis of journal s format, for example, the name of journals should be abbreviated.
We formatted the citations according to the journal style.
- DOI of all references and articles should be added in References.
We added DOI for all references except one reference (Nocard [16]) whose DOI was not found.
- Please, delete Salix in keywords, and write the full name of PGPR in the keywords section.
We removed Salix and added the full name of PGPR.
- If the article has two corresponding article according to what authors have mentioned by adding two emails, ONE STAR ( * ) should be added for the second corresponding author as well.
We added star for Dr. Alotaibi
After these two Minor Revision, the article can be accepted for publication.
Reviewer 3 Report
Comments and Suggestions for Authors
The scope of the revised manuscript is very different now that this is a description of a new species in the Nocardia genus. New text has been added, minor corrections have been made, but the content is still much the same as before.
THE AUTHORS’ RESPONSES TO THE FIRST REVIEW CYCLE ARE NOT SATISFACTORY.
a. As pointed out in item 4 of the previous review cycle, what is the protein that the gene nah1 encodes?
b. As pointed out in item 7 of the previous review cycle, the codes +, ++, and +++ should be explained. It does not help that the code is already explained in a previous publication. The coding has a dual interpretation because it is also used as a +/- code for plant growth-promoting traits. Avoid this confusion.
c. As pointed out in item 9 of the previous review cycle, the references must have a standard format for journal titles (full vs abbreviated titles) and article titles (upper-case vs. lower-case letters). Journal titles are proper nouns and must be capitalized. Delete ‘The’ if it is in the front of a journal title [20, 22]. Genus and species names must be italicized.
d. As pointed out in item 11 of the previous review cycle, include the gene designations in Table S3. The ‘closest to’ column needs a percent similarity value for each match.
e. As pointed out in item 12 of the previous review cycle, the significance of acid phosphatase needs to be addressed in the manuscript. Why is this specific gene mentioned? The function of this gene is disconnected from hydrocarbon degradation.
NEW COMMENTS FROM THE REVIEWER.
1. The revised sections of the text have typos/misprints. For example, line 167, Illunina; character spacing on 52, 55, 57; bolding on 59; what is PGP on 90?
2. 69-81. Reference [2] makes no mention of WB46.
3. Some nouns in the text are missing indefinite and definite articles (examples: 194, 196, 198, 210).
4. The authors claim hydrocarbon utilization degradation by WB46 but no data are presented. The only reference to hydrocarbon degradation is [12], in which WB46 is one of the 50 isolates tested for various traits including growth on diesel.
5. Additional information is required now that the manuscript describes a new species: a formal description of the new species and the deposition of the culture in a recognized public culture collection.
Author Response
We apologize for the unsatisfactory answers in the first draft. We have now taken the time to thoroughly address all the corrections.
Answers point by point (all charges are shown in red color in the manuscript):
The scope of the revised manuscript is very different now that this is a description of a new species in the Nocardia genus. New text has been added, minor corrections have been made, but the content is still much the same as before.
THE AUTHORS’ RESPONSES TO THE FIRST REVIEW CYCLE ARE NOT SATISFACTORY.
- As pointed out in item 4 of the previous review cycle, what is the protein that the gene nah1 encodes?
The gene nah1 encodes Naphthalene dioxygenase. The manes of enzymes were added in Table 1.
- As pointed out in item 7 of the previous review cycle, the codes +, ++, and +++ should be explained. It does not help that the code is already explained in a previous publication. The coding has a dual interpretation because it is also used as a +/- code for plant growth-promoting traits. Avoid this confusion.
We added explanation of the symbols used in Table 1.
- As pointed out in item 9 of the previous review cycle, the references must have a standard format for journal titles (full vs abbreviated titles) and article titles (upper-case vs. lower-case letters). Journal titles are proper nouns and must be capitalized. Delete ‘The’ if it is in the front of a journal title [20, 22]. Genus and species names must be italicized.
All references were reformatted, DOI were added and genera and species names were italized.
- As pointed out in item 11 of the previous review cycle, include the gene designations in Table S3. The ‘closest to’ column needs a percent similarity value for each match.
Gene designations was added. % identity of sequences was also added.
- As pointed out in item 12 of the previous review cycle, the significance of acid phosphatase needs to be addressed in the manuscript. Why is this specific gene mentioned? The function of this gene is disconnected from hydrocarbon degradation.
The significance of acid phosphates was added to the manuscript.
NEW COMMENTS FROM THE REVIEWER.
- The revised sections of the text have typos/misprints. For example, line 167, Illunina; character spacing on 52, 55, 57; bolding on 59; what is PGP on 90?
Corrected
- 69-81. Reference [2] makes no mention of WB46.
The information on the isolate WB46 in the reference [2] can be found in Supplementary Table S1: https://www.mdpi.com/article/10.3390/plants10101987/s1
- Some nouns in the text are missing indefinite and definite articles (examples: 194, 196, 198, 210).
Corrected
- The authors claim hydrocarbon utilization degradation by WB46 but no data are presented. The only reference to hydrocarbon degradation is [12], in which WB46 is one of the 50 isolates tested for various traits including growth on diesel.
In fact, Table #3 and figure #7 (in Alotaibi, F.; St-Arnaud, M.; Hijri, M. In-Depth Characterization of Plant Growth Promotion Potentials of Selected Alkanes-Degrading Plant Growth-Promoting Bacterial Isolates. Frontiers in Microbiology 2022, 13, 863702, showed the hydrocarbon utilization potential of WB46. This reference is included in the body of the manuscript.
- Additional information is required now that the manuscript describes a new species: a formal description of the new species and the deposition of the culture in a recognized public culture collection.
A description of morphological and physiological features have been included in the manuscript. The strain is in the process of being officially deposited in two International collections.
Reviewer 4 Report
Comments and Suggestions for Authors
This manuscript has been improved a lot. However, some works still need to be added to complete the introduction of the new species.
1. Please add the morphology and physiology of the new species in the Method part.
2. Figure 2 needs to be improved, and the new species need to be named and marked in the figure.
3. A official description of the new species need to be added in the Result part.
4. Phylogenetic Analysis is absent in the Method.
Author Response
Answers point-by-point (changes are shown in red in the manuscript).
This manuscript has been improved a lot. However, some works still need to be added to complete the introduction of the new species.
- Please add the morphology and physiology of the new species in the Method part.
We added a description morphology and physiology in the last section of Mat&Meth.
- Figure 2 needs to be improved, and the new species need to be named and marked in the figure.
The new species is marked in red in the figure.
- A official description of the new species need to be added in the Result part.
Morphological and physiological tests results were added to the manuscript.
- Phylogenetic Analysis is absent in the Method.
The phylogenetic analysis is mentioned in lines 195-204 in the manuscript.